# Statins Neuromuscular Adverse Effects

**DOI:** 10.3390/ijms23158364

**Published:** 2022-07-28

**Authors:** Silvia Attardo, Olimpia Musumeci, Daniele Velardo, Antonio Toscano

**Affiliations:** 1Unit of Neurology and Neuromuscular Disorders, Department of Clinical and Experimental Medicine, University of Messina, 98125 Messina, Italy; attardo.silvia@tiscali.it (S.A.); atoscano@unime.it (A.T.); 2Neuromuscular and Rare Diseases Unit, Department of Neuroscience, Foundation IRCCS Ca’ Granda Ospedale Maggiore Policlinico, 20122 Milan, Italy; daniele.velardo@policlinico.mi.it

**Keywords:** statins and myopathy, muscle adverse effects, neuromuscular complications, peripheral neuropathy, myasthenia

## Abstract

Statins are drugs widely prescribed in high-risk patients for cerebrovascular or cardiovascular diseases and are, usually, safe and well tolerated. However, these drugs sometimes may cause neuromuscular side effects that represent about two-third of all adverse events. Muscle-related adverse events include cramps, myalgia, weakness, immune-mediated necrotizing myopathy and, more rarely, rhabdomyolysis. Moreover, they may lead to peripheral neuropathy and induce or unmask a preexisting neuromuscular junction dysfunction. A clinical follow up of patients assuming statins could reveal early side effects that may cause neuromuscular damage and suggest how to better modulate their use. In fact, statin dechallenge or cessation, or the alternative use of other lipid-lowering agents, can avoid adverse events. This review summarizes the current knowledge on statin-associated neuromuscular adverse effects, diagnosis, and management. It is conceivable that the incidence of neuromuscular complications will increase because, nowadays, use of statins is even more diffused than in the past. On this purpose, it is expected that pharmacogenomic and environmental studies will help to timely predict neuromuscular complications due to statin exposure, leading to a more personalized therapeutic approach.

## 1. Introduction

Statins are drugs frequently prescribed in patients suffering from dyslipidemia and, even, in patients with coronary artery disease, diabetes mellitus, stroke, blood hypertension, and chronic kidney disease with or without dyslipidemia.

Advantages of using statins go beyond the simply reduction of cholesterol level, because of several additional positive effects, the so-called ”pleiotropic effects”, as anti-inflammatory, antioxidant, anti-proliferative, apoptotic, cell cycle regulatory, and immunomodulatory effects [1].

Statins are molecules of fungal origin that, by inhibiting the hydroxymethylglutaryl-CoA (HMG-CoA) reductase enzyme, a key step in the sterol biosynthetic pathway, became powerful cholesterol-lowering medications [2].

Statins can be divided into two groups: lipophilic and hydrophilic (Table 1), both demonstrating the “pleiotropic effects” [3].

Lipophilic statins (lovastatin, simvastatin, fluvastatin, atorvastatin, and pitavastatin) cross cell membranes by passive diffusion and are relatively non-selective for hepatic tissues. Hydrophilic statins (rosuvastatin and pravastatin) are unable to cross cell membranes and therefore require activated carrier-mediated transport. They are more selective for hepatic tissues [3].

Statins may also prevent the formation of isoprenoid intermediaries like isopentenyl-pyrophosphate, dimethylallyl-pyrophosphate, geranyl-pyrophosphate, and farnesyl-pyrophosphate. Isoprenoids play an important role in the post-translational modifications and in multiple signaling molecules of membrane attachment [4].

They are predominantly metabolized by the cytochrome P450 (CYP450) system family of enzymes, composed of over 30 isoenzymes. It is also known that lipophilic drugs are much more susceptible to oxidative metabolism via the CYP450 system.

In fact, it is well known that statins, by metabolizing via the CYP450 system, may determine muscle toxicity. This is because the risk of interactions with several other drugs inhibiting CYP450, notably the CYP3A4 isoform, is increasingly producing an elevation of statins plasma levels [5,6,7] (Table 2).

Notably, among statins, it was observed that pravastatin has a minimal drug interaction since it is primarily eliminated by sulfation and should be considered in patients on immunosuppressive therapy post-transplantation or in HIV-patients treated by protease inhibitors [8].

Statins are usually safe and well tolerated, but still there is a low percentage of patients affected by adverse effects.

In the present review, we focused on articles published in PubMed from 1 January 2000 to 1 November 2021, using the search terms “statin and or “myopathy ”, “adverse events”, “myotoxicity”, “myalgia”, “genetic”, “risk factors”, “neuropathy”, and “autoimmnune disease”. We further evaluated the search list for the most relevant articles in the field.

It has been described that some patients, after the use of statins, have reported neuromuscular symptoms, becoming one of the most important causes of statin discontinuation, that, in the meantime, may determine an increased risk of cardiovascular events [9].

Adverse effects on skeletal muscles occur in approximately 5–10% of patients taking statins [10].

Observational studies estimate that 10–15% of statin users develop statin-related muscular side effects, ranging from myalgia and fatigue to more severe muscle symptoms with significant CK elevations [11].

A recent study of Abed et al. evaluated the incidence of myopathy in patients under statins during an observation period of 12 months, showing that the incidence of statin-induced myopathy was 27.4%, highest with simvastatin and lowest with fluvastatin and rosuvastatin [12].

Statins side effects are likely to occur more frequently in elderly patients, because of multiple comorbidities and use of other drugs that may interact negatively with statins [13].

Neuromuscular manifestations of statin intolerance can be highly variable and include more frequently muscle symptoms (myopathy, myalgia, cramps); neuromuscular junction disorders; or, more rarely, peripheral neuropathies. Moreover, an important condition that may occur in patients taking statins is immune-mediated necrotizing myopathy characterized by autoantibodies directed against 3-hydroxy3-methyl-glutaryl-coenzyme-A reductase (HMGCR). This condition usually does not improve with statin withdrawal but needs specific immunosuppressive therapy.

## 2. Statins and Skeletal Muscle Adverse Effects

Statin intolerance is most frequently associated with a wide range of side effects in the skeletal muscle, the so-called “Statin-Associated Muscle Symptoms” (SAMS). SAMS are quite difficult to be diagnosed and managed because there are no validated biomarkers or tests that can be used to confirm their presence, but also because muscle symptoms could originate from other comorbidities [14].

However, a study carried out by Camerino et al., showed that ClC-1 chloride channel protein is reduced in patients with SAMS, and this is often associated with alterations of the electromyographic recordings, although patients may have a statin-induced myotoxicity occurring as muscle necrosis due to statin exposure and manifesting with increased CK levels.

ClC-1 plays an important role in sarcolemma hyperexcitability and its reduction in skeletal muscles of patients using statins may help to evaluate the risk of myopathy [15].

The risk of SAMS is higher with lipophilic statins such as simvastatin, atorvastatin, and lovastatin, because of their ability to not selectively diffuse into extrahepatic tissues as skeletal muscles. On the other hand, hydrophilic statins, such as pravastatin and fluvastatin, are associated to a lower risk of SAMS [16].

Myotoxicity may also depend on the statin’s metabolic profile. In fact, according to Catapano et al. co-administration of drugs may inhibit the cytochrome P450 (CYP) enzymes, responsible for metabolizing statins, or interact with the organic anion-transporting polypeptides (OATPs), responsible for statin uptake into hepatocytes, substantially increasing the risk of developing myotoxicity [17].

Several statins widely used in clinical practice, such as atorvastatin, simvastatin, and lovastatin (and previously cerivastatin, now off the market because of very high incidence of rhabdomyolysis) [18], are metabolized by the cytochrome P450 (CYP)3A4 pathway. Some statins are metabolized by different enzymes such as fluvastatin by CYP2C8 or pravastatin and rosuvastatin metabolized by diverse pathways [19].

According to Ramkumar et al., atorvastatin and rosuvastatin are both good choices in order to reduce the occurrence of statin myopathy, because they have long half-lives enabling alternate day or twice weekly dosing strategy [8].

The most important risk factors of SAMS are advanced age, female gender, Asian ethnicity, drugs altering statin plasma levels, excessive physical activity, muscle, liver or chronic kidney diseases, uncontrolled hypothyroidism, abdominal obesity and metabolic syndrome, and vitamin D deficiency (Table 3).

Typically, SAMS manifests with different muscle symptoms occurring after statin treatment (with or without elevations of serum creatine kinase) that might resolve after its interruption.

SAMS has a highly variable clinical presentation ranging from a myopathic pattern, characterized by muscle tenderness, cramping and muscle aches, weakness and increased CK level (even 10 times higher the upper normal limit), to rhabdomyolysis [20]. Myopathy usually appears in patients who receive high doses of statins, especially when taking simvastatin 80 mg daily, which lead to higher plasma levels of active statins metabolites, especially in the first year of treatment or after having increased the dosage. Muscle disorders are often reversible after statin withdrawal [13].

Many questionnaires have been proposed to detect statin-associated muscle symptoms in clinical practice and to optimize treatment for patients experiencing muscle symptoms.

The most known questionnaire is the “Statin-Associated Muscle Symptom Clinical Index (SAMS-CI)”, which includes four separate ratings: the first regarding the location and pattern of the muscle symptoms, while the other three consider timing of symptoms relative to start, stop (dechallenge), and rechallenge with statins.

The total score of SAMS-CI allows to establish the likelihood that patients’ muscle symptoms are due to statin use [21].

Other questionnaires available are the Patient and Provider Assessment of Lipid Management (PALM) and the Statin Experience Assessment Questionnaire (SEAQ). This is a self-evaluation composed of nine questions: seven regarding muscular symptoms, evaluated by patients with a scale from 0 to 10, and the other two regarding the possible discontinuation of therapy [22].

The most important clinical manifestations of SAMS are muscle weakness and muscle pain, mainly interesting large muscle groups (thighs, buttocks, calves, and back muscles) and, usually, manifesting early, but also after several years of treatment [23].

Rhabdomyolysis is a condition characterized by muscle necrosis, which causes release of myoglobin into the bloodstream, showing a very much elevated CK level (even 40 times higher than normal). This condition may require hospitalization, because it can induce myoglobinuria and acute renal failure [13].

The risk of statin-induced rhabdomyolysis is significantly increased when other drugs such as fibrates, cyclosporine, macrolide antibiotics, and azole antifungals are concomitantly administered [24]. Law et al. indicated that the incidence of rhabdomyolysis may be higher in patients taking lovastatin, simvastatin, and atorvastatin, or using other inhibiting drugs (see Table 1), because of their metabolism regulated by CYP3A4 isoform of cytochrome P450. On the other hand, fluvastatin (oxidized by CYP 2A9), pravastatin (not oxidized by CYP 2A9), or rosuvastatin are less harmful because they are related to different metabolic pathways. Rate of risk of rhabdomyolysis, revealed by FDA Adverse Effects Reporting System (AERS), is approximately four times higher in monotherapy with lovastatin, simvastatin, and atorvastatin, than in pravastatin, rosuvastatin, or fluvastatin. It has also been evidenced that the incidence of rhabdomyolysis is approximately 10 times higher when fibrates are contemporarily utilized [25].

## 3. Pathogenesis of SAMS

It is important to outline that the precise mechanisms producing SAMS are still not clearly defined.

According to the literature data, mitochondrial dysfunction could be the primary mechanism underlying statin myopathy.

Mitochondrial dysfunction can be defined as a decrease in the ability of mitochondria to synthesize high-energy compounds such as adenosine 5′ triphosphate and because of a suboptimal electron transfer rate across the respiratory chain complexes.

The exact mechanism through which mitochondrial dysfunction may cause SAMS remains unclear; however, it can be associated with a reduction of CoQ10 levels [26].

In the mitochondrial metabolism, Coenzyme Q10 (CoQ10), also known as ubiquinone, is one of the end products of the mevalonate pathway. Statins use lead to its depletion in a dose-dependent manner and it could be associated with an elevated risk of myopathy [20].

Statins block the production of farnesyl pyrophosphate, an intermediate of the mevalonate pathway responsible for CoQ10 production, suggesting that the reduction in plasma CoQ10 concentrations may contribute to SAMS. In addition, patients treated with statins frequently have a decreased muscle CoQ10 concentration, suggesting that statins might impair muscle mitochondrial function [27].

For this reason, CoQ10 has been considered as a form of adjuvant therapy for SAMS treatment but data evaluating the efficacy of CoQ10 supplementation have been controversial because not all the studies have proved that CoQ10 supply mitigates muscular complaints [28].

Other possible mechanisms explaining statin myopathy are reduced sarcolemmal cholesterol and isoprenoids involved in muscle fiber apoptosis [29].

During statins administration, apoptosis of myofibers may be a pathogenic mechanism that is induced by isoprenoid depletion, leading to low levels of protein geranylgeranylation and/or farnesylasion [30].

## 4. Genetic Origin of SAMS

The genetic basis of SAMS has been investigated extensively. In fact, genetic background also plays an important role, increasing the susceptibility to SAMS, particularly in subjects with additional risks (Table 3).

Polymorphisms of the SLCO1B1 gene are the best-known pharmacokinetic changes in statin-associated myopathy. The *SLCO1B1* gene product is responsible for hepatic uptake of statins.

A coding variant (p.Val174Ala, rs4149056) in the *SLCO1B1* gene was found to be significantly associated with myopathy, making this the most validated genetic basis for SAMS.

rs4149056 is a non-synonymous missense variant in exon 5 of the *SLCO1B1* gene, altering a valine amin oacid with alanine at position 174. This variant has a strong association with simvastatin, and for this reason, patients with rs4149056, taking this kind of statin, should assume a lower dose of simvastatin or other statins with a narrow monitoring of creatine kinase (CK levels) [31,32].

It seems that patients with neuromuscular disorders and genetic myopathies, such as Pompe disease, Kennedy disease, McArdle disease, amyotrophic lateral sclerosis, myotonic dystrophy 1 and 2, and muscle phosphorylase b kinase deficiency, are more predisposed to develop SAMS.

There are other genes related with SAMS phenotypes such as *COQ2*, *HTR7*, *RYR1*, *GATM*, *CYP3A4*, *CYP2D6*, *ABCC2*, *RYR2*, *CLCN1*, *VDR,* and *ABCG2.* It was found that also the *LPIN1* gene mutation may lead to myopathy and rhabdomyolysis in children [31].

Moreover, it was found that pathogenic variants in metabolic genes including *RYR1* and *CACNA1S* could be associated to statin muscle symptoms. Mutations or variants in *RYR1* and *CACNA1S* genes have been found to be more frequent in statin myopathy patients than controls [33].

Another genetic factor linked to SAMS susceptibility is a variation in the CoQ2 gene encoding for parahydroxybenzoate-polyprenyl-transferase enzyme. Some variants of this gene are associated with CoQ10 deficiency and skeletal muscle drug transporters expression, leading to statin muscle intolerance [34].

## 5. Classification of Statin-Related Myopathic Phenotypes

A classification of statin-related muscular phenotypes that include seven different categories was proposed: SRM0 (asymptomatic CK elevation); SRM1 (tolerable myalgia with no CK elevation); SRM2 (intolerable myalgia with CK elevation); SRM3—myopathy; SRM4 (severe myopathy); SRM5 (rhabdomyolysis); and SRM6 (immune-mediated necrotizing myopathy).

According to this classification, Turner et al. showed that symptoms more frequently involve lower limbs muscles (thighs, calves), although back, neck, and shoulder muscles may also be involved [34,35].

According to Kia et al., in about 5% of patients on statins increased CK levels may occur, whose normalization usually requires at least several days [36].

## 6. Statins and Peripheral Neuropathy

It has been observed that statins may also cause a peripheral neuropathy, although this risk is quite low with an incidence of approximately 12 per 100,000 person per year, or with a prevalence of 6 per 10,000 persons, or 1 in 10,000 patients treated for 1 year [37,38].

Patients on statin therapy may develop a peripheral neuropathy, complaining of numbness, tingling, pain, and tremor at hands or feet, as well as unsteadiness during walking. All these symptoms are usually generated by a long-term therapy (>1 year) [39].

Moreover, the incidence of polyneuropathy has been reported more frequently with atorvastatin than with fluvastatin [40].

From the diagnostic point of view, neurophysiological investigations, based on nerve conduction studies and needle electromyography, may be useful for an appropriate diagnosis of peripheral neuropathy by detecting motor and sensory nerves involvement [41].

Many studies found that the duration of statin therapy is a significant risk factor in the development of sensory neuropathy.

A study carried out among 30 statin-treated patients versus patients without statins, showed a decreased vibration perception in the treated group, which is suggestive of peripheral sensory neuropathy [42].

According to Emad et al., statins may lead to peripheral nerve alterations with an axonal involvement but without clinical manifestations [43].

A study carried out by Otruba demonstrated that long-term treatment with statins caused a clinically silent, but electrophysiologically defined, damage of peripheral nerves.

The study included 42 patients (23 males, 19 females, mean age 51.9 and 52.3 years) with a diagnosis of combined hyperlipidemia in treatment with simvastatin of 20 mg daily. Patients were followed for 24 months; none of the patients reported subjective symptoms typical for polyneuropathy. However, electrophysiological examination of lower-limb peripheral nerves demonstrated statistically significant prolongation of F-wave mean latency on peroneal and tibial nerves (*p* < 0.0001, paired *t*-test). A control group of 50 patients with combined hyperlipidemia without statin treatment did not show alterations over the same time interval [44].

Gurha et al. found that the reduction of CoQ10 levels in blood might have been responsible for the onset of a peripheral neuropathy, demonstrated by nerve conduction alterations [45].

On the other hand, a Danish registry-based study found that statin use was not associated with an increased risk of idiopathic polyneuropathy [46].

One can conclude that the exact mechanism by which statins can cause neuropathy is still undefined. A possible theory is that statins, inhibiting cholesterol synthesis, may impair the function of nerve membranes [41].

According to Lehrer et al., statins combined with niacin (vitamin B3) may reduce the risk of peripheral neuropathy. Among the B-group vitamins, niacin has long been recognized as a key mediator of neuronal development and survival and, for this reason, may be useful for the treatment of neuropathy [47].

However, early detection of peripheral neuropathy and changing hypercholesterolemia treatment may prevent permanent nerve damage [39].

In conclusion, although some observational studies have suggested a possible association between statins and peripheral neuropathies, results are quite variable. In fact, there is no conclusive evidence for a constant causal relationship between statin treatment and peripheral neuropathy [48].

## 7. Statins and Autoimmune Illnesses

### 7.1. Statins and Myasthenia Gravis

A rare side effect of statins is the possible induction of autoimmune illnesses such as dermatomyositis, polymyositis, immune-mediated necrotizing myopathy (IMNM), and myasthenia gravis (MG).

Statins are included in the list of drugs that may promote or exacerbate MG presentation, although this specific influence is unknown [49].

MG is an autoimmune disorder characterized by muscle weakness due to an altered transmission at the neuromuscular junction, with autoantibodies against acetylcholine receptor (AChRAb) or, less commonly, against a muscle-specific tyrosine kinase (MuSKAb) [50].

Shin et al. carried out a study including 185 MG patients, of whom 54 (31%) patients were on statins. In this study, MG worsening was observed in 6 patients and occurred in all MG forms. In addition, all types of statins were associated with MG worsening. This study has also shown that the most recurrent symptom in MG worsening is an oculo-bulbar weakness: five patients experienced bulbar symptoms, while only one patient reported limb weakness [51].

Keogh et al. described a case report of a 60-year-old Caucasian man of British origin, with a diagnosis of hyperlipidemia treated with simvastatin who showed dysarthria and dysphagia associated with proximal muscle weakness; myalgia; and, even, a very elevated CK level (2599 U/L). Edrophonium testing demonstrated a significant transient improvement of dysarthria, and a diagnosis of myasthenia gravis with positive anti-acetylcholine receptor antibodies was confirmed [49].

Cartwright et al. reported a patient with MG related to statin use. The patient, a 55-year-old man without previous history of myasthenia, developed recurrent dysarthria after having started a treatment with atorvastatin. He responded to pyridostigmine, showing abnormal decremental responses on repetitive nerve stimulation and transiently elevated AChR antibodies [52].

Purvin et al. reported another four cases of statin-associated MG. These patients had abnormal electrodiagnostic testing and three out of four showed positive AChR antibodies. Two patients improved without therapy, while the other two only after corticosteroid therapy (prednisone) [53].

The association between statins and MG is not clear and it may be due to immunomodulatory effects of statins on T and B cells [51].

Statins may determine the inhibition of T-cell activation, as well as B-cell proliferation, differentiation, and induction of B-cell apoptosis in mice. In humans, statins may inhibit human B-lymphocyte activation and MHC class II antigen presentation [54]. As a result, this may lead to an increase of antibody-mediated humoral immunity [55].

On the other hand, statin may interfere with therapy for MG. It has been reported that statins impair the effects of rituximab by inducing conformational changes of CD20, which are to be take in account in patients with refractory MG who benefit from this treatment [56,57].

Statins have the same indications in patients with MG as in the whole population: a patient with myasthenia and dyslipidemia should take statins. However, it is correct to inform patients that statins could worsen myasthenic symptoms in order to withdraw statins if this happens [58].

### 7.2. Immune-Mediated Necrotizing Myopathy Statin-Related

A rare muscle condition associated with statin treatment is the immune-mediated necrotizing myopathy (IMNM), considered as a very severe autoimmune myopathy that may present with three distinct clinical subtypes related to the presence of different kinds of pathogenic conditions such as antibodies anti-signal recognition particle (SRP), antibodies anti-3-hydroxy-3-methylglutarylCoA reductase (HMGCR), or absence of both of them (antibody-negative IMNM). According to Thomas et al., around 60% of cases are associated with antibodies to SRP or HMGCR, while the others are seronegative [59].

There is a strong association between IMNM in patients exposed to statins and anti-HMGCR antibodies: antibody levels correlate with the severity of myopathy [60].

A diagnosis of IMNM should be considered in patients with suspected statin-induced myopathy in whom symptoms persist or worsen despite discontinuation of drug exposure [61].

As regards as the pathophysiology of anti-HMGCR myopathy, it is well known that statins increase HMGCR expression in muscle and other tissues. Elevated HMGCR protein levels could induce an aberrant processing by antigen-presenting cells with the production of cryptic epitopes that might determinate an autoimmune response. Statin binding to HMGCR could cause conformational changes of the protein, altering its cleavage by antigen-presenting cells with the generation of cryptic epitopes. Somatic or genomic sequence variants in HMGCR could positively influence its immunogenicity as documented for other autoimmune diseases such as scleroderma.

Therefore, statins may have immunomodulatory effects on the immune system changing autoantigen processing [60].

The immunogenic background also plays an important role for the predisposition to develop anti-HMGCR myopathy.

Several studies have shown a strong association of anti-HMGCR myopathy and HLA-DRB1*11:01 allele. This was originally identified in nearly 70% of patients and now has been validated in multiple cohorts of patients with different racial and ethnic origins as an immunogenetic risk factor. A second allele, HLA-DRB1*07:01, has been reported in pediatric patients [60,62].

Among patients using statins, the estimated incidence rate of IMNM is 2-3/100,000 patients with an increased risk among patients over 50 years of age [63].

Muscle symptoms typically occur within a month after statins start or after having increased the dosage [64].

IMNM is characterized by limb-girdle progressive muscle weakness, mainly involving posterior and medial thigh and gluteal compartments. As regards the classical adult onset of anti-HMGCR myopathy, fatigue and myalgia are also present in 20–60% of patients. Dysphagia is reported in 16–30% of patients, whereas others have shown truncal weakness as a unique clinical feature. Non-specific systemic and extramuscular symptoms (e.g., rash, arthritis, Raynaud’s phenomenon) are uncommon. As other autoimmune diseases, anti-HMGCR myopathy seems to be slightly more predominant in females. Elevated CK values are between 10 and 100 times the upper limit (2000 and 20,000 IU/L) [61,65].

The main diagnostic tools of IMNM are EMG, muscle biopsy, MRI, and anti-HMGCR antibodies.

EMG is usually characterized by the presence of fibrillation and sharp waves.

Muscle biopsy is the gold standard for diagnosis, usually showing a “necrotizing myopathy” characterized by morphological aspects as large myofiber necrosis with regeneration (Figure 1 and Figure 2) [66,67].

Immune-mediated features, such as endothelial membrane attack complex (MAC) deposition in non-necrotic fibers and MHC class I expression, are additional pathogenic features of this condition [68,69].

Although studies have reported variability in biopsy samples, macrophages are often reported as the most common infiltrating cells, while CD4+ helper T cells and CD8+ cytotoxic T cells are less commonly found in statin-associated IMNM. However, Zaki et al. reported a case of statin-associated IMNM with atypical biopsy features such as predominant inflammatory aspects represented by CD4+ and CD8+ T cells, in addition to the more typically expected CD68+ macrophages [70].

Apart from muscle biopsy, skeletal muscle magnetic resonance imaging (MRI) is a fundamental tool for IMNM diagnosis and follow-up. In a study carried out by Villa et al., MRI imaging showed the involvement of the dorsal muscle groups of both the thighs and in association with more extensive edema and a trend towards fatty muscle replacement. Moreover, a pattern involving the medial gastrocnemius, triceps, and deltoid, followed by infraspinatus and subscapularis with an inflammation of the subcutaneous tissues and of muscolaris fasciae of both arms and legs, suggestive of a systemic inflammatory response, was found [63].

Muscle MRI may show T1 hyperintensity especially in the posterior thigh STIR signal is increased and may be asymmetric. Liang et al. carried out a study in which they enrolled five patients with anti-HMGCR myopathy; three were pediatric and two were adults. They underwent muscle MRI, and it was found that adductors were earlier affected, while legs were relatively spared with the highest degree of involvement of the medial head of gastrocnemius. At the upper extremities, the biceps brachii was the most severely involved, followed by the triceps [71].

It is important to underline that in patients taking statins with muscle weakness and CK elevation not resolving after discontinuation of statins, anti-HMGCR antibodies should be tested in order to define the diagnosis.

### 7.3. Treatment of IMNM

An important clue regarding this condition is that statin withdrawal does not usually improve patient symptoms, but immunosuppressive treatment is required to obtain a good clinical response [68].

However, the first therapeutic step in IMNM is to stop the statin; if the patient symptoms do not improve and progress, steroids such as prednisone or methylprednisolone are the following line of treatment.

Initially, prednisone is used at 1 mg per kilogram of body weight per day with a maximum of 80 mg daily.

The second step is immunosuppressive therapy such as methotrexate, azathioprine, and mycophenolate mofetil. For those who have moderate symptoms, immunosuppressive medications can be also used as initial therapy in combination with prednisone. If patients do not respond to combination therapy after 8–12 weeks, intravenous immunoglobulin (IVIG) or rituximab may be administered.

However, in severe cases, IVIG is considered as the first line of treatment. The duration of treatment is determined by the patient symptoms, even checking CK levels [72].

Despite the use of immunosuppressive medications such as prednisolone and methotrexate, plasma exchange, and/or intravenous immunoglobulin, some patients do not respond at all. Recently, it was found that rituximab is a very effective immunosuppressive treatment for patients with refractory IMNM [73,74,75].

## 8. Management of Statin Adverse Effects

Currently, there are no guidelines to help physicians to determine whether neuromuscular effects are the result of statin myotoxicity or an unmasking underlying neuromuscular disorder [76].

Adverse effects may induce patients to discontinue statins, reducing cardiovascular benefits [77].

The National Lipid Association (NLA) defines statin intolerance as “the inability to tolerate at least two statins, one at the lowest starting daily dose and another at any daily dose, either due to objectionable symptoms (real or perceived) or abnormal laboratory analysis, temporally related to statin treatment, reversible upon statin discontinuation, reproducible by rechallenge (restarting medication), and excluding other known factors” [9].

Several different approaches to statin-intolerant patients have been suggested, but an evidence-based consensus is difficult to be reached due to the lack of controlled trials.

In order to check the side effect of statins on muscle, the ESC and AHA/ACC guidelines recommend testing basal CK levels before prescribing a statin and recommend testing only if symptoms occur. The criteria of severe myopathy are simply defined as a CK increase to 10 times or more than the upper normal limit along with symptoms. If the symptoms are not severe and tolerable, they recommend reassessing and conducting a statin rechallenge [78,79]. The first step in managing intolerant patients is to determine whether neuromuscular adverse events are indeed related to statin therapy.

There are many options to be considered step-by-step: (a) reduction of the statin dose (dechallenge), (b) switching to a different statin, (c) using intermittent dosages (alternate-day therapy) [80].

The strategy for managing a patient with possible statin-induced myopathy was proposed by Oddis et al. (see Figure 3 modified) [81].

For patients with true statin intolerance, the use of a non-statin lipid lowering agents (such as ezetimibe, colesevelam and red yeast rice) is a valid alternative [80].

PCSK9 inhibitors (alirocumab and evolocumab) have been approved in the USA and in Europe for patients who are statin intolerant. A trial comparing alirocumab to ezetimibe in patients intolerant to two or more statins demonstrated that alirocumab was associated with a lower rate of muscle symptoms compared with atorvastatin [9].

As regards treatment, since reduction in mitochondrial CoQ10 has been proposed as mechanism for SAMS, a possible treatment that may reduce muscle toxicity due to statins is the use of oral formulations of CoQ10 with questionable efficacy [82]. However, a recent study by Chen et al. compared 64 CoQ10 users versus 447 non-CoQ10 users with SAMS and showed that the use of CoQ10 was not associated with a significant improvement of statin-associated muscle symptoms [83].

Since low vitamin D levels (below 80 nmol/L) may be considered a risk factor for SAMS, vitamin D supplementation is a possible option for the prevention of statin intolerance.

In few patients with statin-related myalgia, intravenous administration of vitamin D (50,000 IU once weekly for several weeks) improved symptoms [84,85,86].

According to Balestrino et al., creatine supplementation prevents statin myopathy in statin-intolerant patients. In fact, it has been suggested that statin-induced creatine deficiency might be a major cause of muscle toxicity, due to the inhibition of guanidinoacetate methyl transferase (GAMT), an enzyme that plays an important role in the synthesis of creatine [87].

## 9. Conclusions

The aim of this review is to supply a more complete as possible panorama of the neuromuscular adverse effects in patients assuming statins. Other previous assessments were mainly oriented towards a specific kind of disorder (i.e., myopathy, myasthenia, neuropathies) [39,55].

Statins can cause neuromuscular side effects that represent about two-thirds of all adverse events. The risk of muscle toxicity is higher when using lipophilic statins.

However, statins may also trigger autoimmune disorders: In fact, they may exacerbate or, even, cause myasthenia gravis or IMNM, very severe inflammatory myopathy, that improves with an immunosuppressive therapy.

Moreover, a causal relationship between statins and peripheral nerve damage is quite debated and controversial.

An early recognition of statin-associated side effects is necessary in order to avoid acute or chronic neuromuscular damage and to better modulate statin use. In fact, statin dechallenge or cessation, or alternative administration, using other lipid-lowering agents, may avoid neuromuscular adverse events. The incidence of neuromuscular complications is expected to increase because of an even larger diffusion of statins use. For this reason, it is predictable that pharmacogenomic and environmental studies will help to increase awareness on neuromuscular complications due to statin exposure, leading to a more personalized therapeutic approach [88].

## Figures and Tables

**Figure 1 ijms-23-08364-f001:**
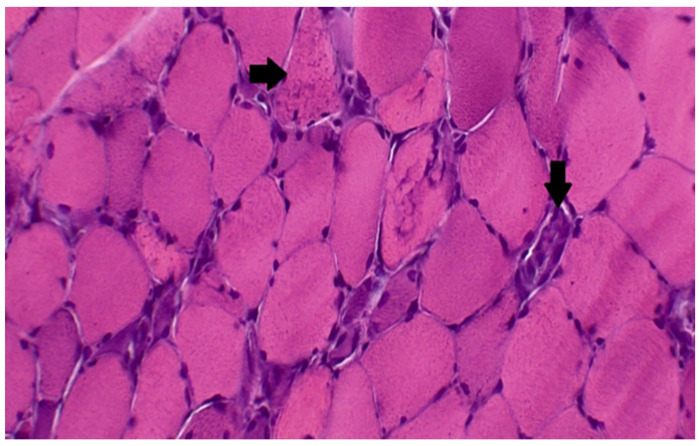
Muscle biopsy: hematoxylin and eosin staining demonstrating the loss of normal muscle architecture, and extensive myofiber necrosis (right arrow). (from Madgula, A. S. et al., 2020 *Cureus*) [66].

**Figure 2 ijms-23-08364-f002:**
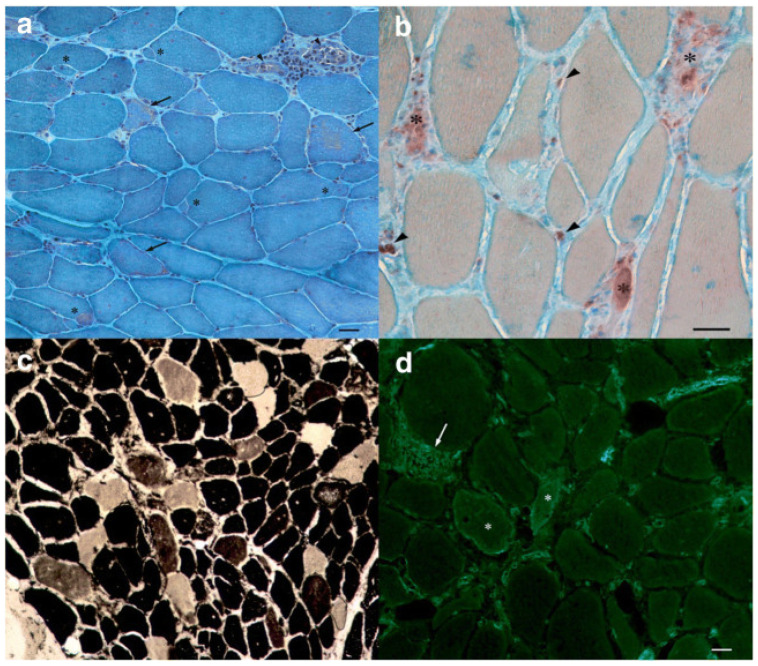
Muscle histology (magnification ×200, scale bar 50 μm). (**a**) Gomori trichrome stain showing increased fiber diameter variability, pale degenerating fibers (arrows). (**b**) Acid phosphatase stain showing phagocytes invading degenerating fibers (asterisks) and endomysial inflammatory cells (arrowheads). (**c**) ATPase pH 4.6 stain showing the normal checkerboard pattern of type 1 (dark) and type 2 (2a pale; 2b brownish) fibers. (**d**) MHC-1 immunofluorescence showing cytoplasmic and sarcolemmal positivity in some necrotic fibers (asterisks) and membrane staining of inflammatory cells (arrow) (from Barp A. et al., 2021 Neurol Sci) [67].

**Figure 3 ijms-23-08364-f003:**
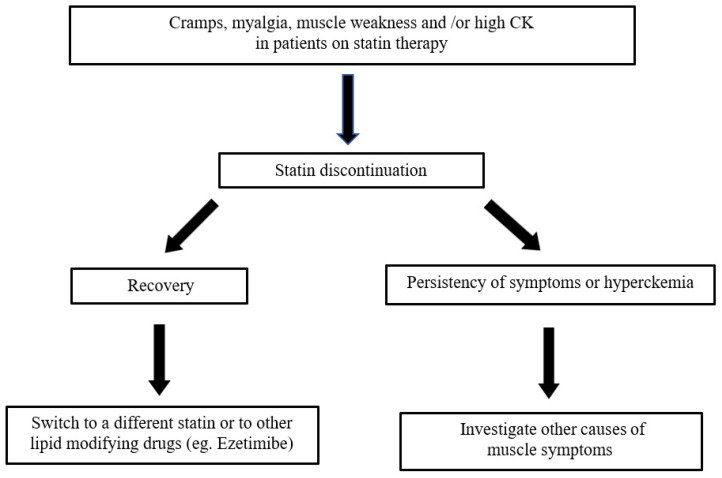
Strategy for statin-associated muscle symptoms.

**Table 1 ijms-23-08364-t001:** Statins subtypes.

Lypophilic Statins	Hydrophilic Statins
Atorvastatin	Rosuvastatin
Fluvastatin	Pravastatin
Simvastatin	
Pitavastatin	
Lovastatin	

**Table 2 ijms-23-08364-t002:** Drugs inhibiting CYP450 likely interfering with statins.

**Inhibitors of CYP3A4** **(atorvastatin, simvastatin, lovastatin)**	azole antifungals (ketoconazole, fluconazole), macrolide antibiotics (erythromycin, clarithromycin, troleandomycin, telithromycin), antidepressants (fluvoxamine, fluoxetine, sertraline, nefazodone) calcium channel antagonists (amlodipine, verapamil, diltiazem, mibefradil), protease inhibitors (indinavir, ritonavir), immunosuppresants (cyclosporine, erlotinib), clopidogrel, grapefruit juice
**Inhibitors CYP2C9** **(fluvastatin, rosuvastatin)**	azole antifungals (ketoconazole, fluconazole, itraconazole), fluoxetine, amiodarone, warfarin
**Inhibitors of organic anion transporting peptide 1B1**	gemfibrozil

**Table 3 ijms-23-08364-t003:** Risk factors associated with SAMS.

Family or Personal History of Intolerance to Statins	References
Advanced age	[9,10,13,16]
Female gender	[9,16]
Asian ethnicity	[9,16]
Drugs altering statin plasma levels	[13,16]
Excessive physical activity	[9,16]
Muscle diseases	[9,16]
Liver diseases	[9,16]
Chronic kidney diseases	[9,16]
Uncontrolled hypothyroidism	[9,16]
Abdominal obesity and metabolic syndrome	[9,16]
Vitamin D deficiency	[9,16]

## Data Availability

Not applicable.

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
