# Peer review of "Statins Neuromuscular Adverse Effects"

_ijms, 2022, doi:10.3390/ijms23158364_

Round 1

Reviewer 1 Report

Authors summarized the current knowledge regarding statin-associated neuromuscular side effects, diagnosis and treatment.

1. I think that in Table 2 some more drugs should be mentioned.
2. Please consider citing: 
- Winiarska M, Bil J, Wilczek E, Wilczynski GM, Lekka M, Engelberts PJ, Mackus WJ, Gorska E, Bojarski L, Stoklosa T, Nowis D, Kurzaj Z, Makowski M, Glodkowska E, Issat T, Mrowka P, Lasek W, Dabrowska-Iwanicka A, Basak GW, Wasik M, Warzocha K, Sinski M, Gaciong Z, Jakobisiak M, Parren PW, Golab J. Statins impair antitumor effects of rituximab by inducing conformational changes of CD20. PLoS Med. 2008 Mar 25;5(3):e64. 
- Winiarska M, Bil J, Wilczek E, Wilczynski GM, Lekka M, Engelberts PJ, Mackus WJ, Gorska E, Bojarski L, Stoklosa T, Nowis D, Kurzaj Z, Makowski M, Glodkowska E, Issat T, Mrowka P, Lasek W, Dabrowska-Iwanicka A, Basak GW, Wasik M, Warzocha K, Sinski M, Gaciong Z, Jakobisiak M, Parren PW, Golab J. Statins impair antitumor effects of rituximab by inducing conformational changes of CD20. PLoS Med. 2008 Mar 25;5(3):e64.
3. Table 3 - Please add here more data or delete the 2-row table. Please also explain IMNM under the table
4. Table 4 - maybe you could add OR/HR ranges
5. Please refer to any ESC or AHA/ACC cardiology strategy to manage CK increase on statin therapy.

Author Response

Q I think that in Table 2 some more drugs should be mentioned.

A According to the reviewer suggestions, we have modified Table 1 grouping the different types of drugs and including some more medications reported in the literature

Q Please consider citing: 
- Winiarska M, Bil J, Wilczek E, Wilczynski GM, Lekka M, Engelberts PJ, Mackus WJ, Gorska E, Bojarski L, Stoklosa T, Nowis D, Kurzaj Z, Makowski M, Glodkowska E, Issat T, Mrowka P, Lasek W, Dabrowska-Iwanicka A, Basak GW, Wasik M, Warzocha K, Sinski M, Gaciong Z, Jakobisiak M, Parren PW, Golab J. Statins impair antitumor effects of rituximab by inducing conformational changes of CD20. PLoS Med. 2008 Mar 25;5(3):e64. 
A We have included this reference in the section MG and statins (7.1), considering that rituximab has demonstrated to be quite efficacious in refractory MG

Q Table 3 - Please add here more data or delete the 2-row table. Please also explain IMNM under the table
A We have decided to eliminate Table 3 including its content in the text

Q Table 4 - maybe you could add OR/HR ranges
A Table 4 is now Table 3: we have looked at  the literature searching for studies analysing the OR/HR for clinical risk factors and we found data only for  someof the risks or for single type of statins. For this reason, we have decided to add a column in the table 3 adding some references for each kind of risk

Q Please refer to any ESC or AHA/ACC cardiology strategy to manage CK increase on statin therapy.

A We have added to the section “Management of statin adverse effects”, the guidelines from ESC or AHA/ACC reporting the last European and American guidelines. We have included both of them in the reference list

Reviewer 2 Report

I congratulate the authors on a very interesting work: "Statins neuromuscular adverse effects "

This is a very important issue. The use of statins is common. They have a good healing effect not only in heart and vascular diseases. The manuscript systematizes knowledge about statins and their side effects. It also shows ways to avoid these side effects. This is very important because statins are used by a large group of patients. I propose to mark in figure 1 its components A, B, C.

Author Response

Q I congratulate the authors on a very interesting work: "Statins neuromuscular adverse effects "

This is a very important issue. The use of statins is common. They have a good healing effect not only in heart and vascular diseases. The manuscript systematizes knowledge about statins and their side effects. It also shows ways to avoid these side effects. This is very important because statins are used by a large group of patients. I propose to mark in figure 1 its components A, B, C.

A We have modified the figure 1 including some pictures published by Barp et al (2021) and by Magdula et al (2020). Both references are now in the list

Reviewer 3 Report

The  aim of this review is to summarize the current knowledge regarding statin-related neuromuscular side effects, diagnosis and treatment. The effort to put this review together should be commended. However there is lack of information what databases were used to search the papers and used search term strategies.

I think the content of this article can attract readers' interest. The logic of the article is clear and the sentences are smooth.

I have the following comments:

1) Section 1:

Table 1: Present lipophilic statins in regular font instead of bold, delete the dash in the tittle.

Table 2: Delete the dash in the tittle.

Table 3: Terms like myopathy, myalgia etc should be descripted and their histopathological characterization also.

2) Section 2:

In this section, the authors need to describe term myotoxicity.

Table 4 and 5:  Delete the dash in the tittle.

Table 4: The contents should be divided for SAMS less and more likely risk factors.

3) Section 4:

There are more genes related with SAMS phenotypes. It needs revision.

Names of genes should be writen italic (192, 212, 215, 216, 219 line).

4) Section 5:

Definition of ULN is needed.

5) Section 7:

I would suggest to reorganize this section. To my opinion, the authors should follow this plan.

Section 7 title: Statins and autoimmune illnesses. Next, 7.1 Statins and myasthenia gravis.

Then, 7.2 Immune-mediated necrotizing myopathy statin-related. Finally, 7.3,  Treatment of IMNM.

Line 387:  Should be Figure 1 (A, B and C).

In Figure 1, indicators A, B and C are omitted. Figures should have arrows indicating necrosis of the muscle fibers. Unnecessary underlining of the sentence in the description of Figure 1.

6) Section 9

Line 466: Figure 2 should be referenced, not Figure 1.

Author Response

Q The aim of this review is to summarize the current knowledge regarding statin-related neuromuscular side effects, diagnosis and treatment. The effort to put this review together should be commended. However there is lack of information what databases were used to search the papers and used search term strategies.

 I think the content of this article can attract readers' interest. The logic of the article is clear and the sentences are smooth.

A According to your suggestion, we have included the search databases used with the keywords and the search lenght

Q I have the following comments:

1) Section 1:

Table 1: Present lipophilic statins in regular font instead of bold, delete the dash in the tittle.

Table 2: Delete the dash in the title.

Table 3: Terms like myopathy, myalgia etc should be descripted and their histopathological characterization also.

A We have modified the table 1 and 2 but we have decided to delete table 3 according to the reviewer 1 suggestion

Q 2) Section 2:

In this section, the authors need to describe term myotoxicity.

Table 4 and 5: Delete the dash in the tittle.

Table 4: The contents should be divided for SAMS less and more likely risk factors.

A We defined myotoxitcity as muscle necrosis induced by statin exposure and manifesting with hyperCKemia.

We updated the table 4 and 5

Q Section 4:

There are more genes related with SAMS phenotypes. It needs revision.

Names of genes should be writen italic (192, 212, 215, 216, 219 line).

A We have updated the numbers of genes according to some recent references (Turongkaravee S,2021 and Isackson pJ, 2018) reported in the list 

Q Section 5:

Definition of ULN is needed.

A The definition is already reported (Upper limit of normal)

Q Section 7:

I would suggest to reorganize this section. To my opinion, the authors should follow this plan.

Section 7 title: Statins and autoimmune illnesses. Next, 7.1 Statins and myasthenia gravis.

Then, 7.2 Immune-mediated necrotizing myopathy statin-related. Finally, 7.3, Treatment of IMNM.

A We have updated the section according to the reviewer suggestions

Q Line 387: Should be Figure 1 (A, B and C).

In Figure 1, indicators A, B and C are omitted. Figures should have arrows indicating necrosis of the muscle fibers. Unnecessary underlining of the sentence in the description of Figure 1.

A We have modified the figure 1 including some pictures published by Barp et al (2021) and by Magdula et al (2020). Both references are now in the list

Q Section 8

Line 466: Figure 2 should be referenced, not Figure 1.

A We have modified it according to reviewer suggestion

Reviewer 4 Report

The topic submitted has already large reviews published earlier but does cover the latest significant research data to the existing field of research of statins. The article is not very well articulated and needs English language revisions and even formatting of the manuscript as per the MDPI guidelines. The manuscript needs to be checked for proper numbering and citation of references. The introduction needs to be enlarged with more background. Abstract and conclusion should include a sentence proposing the future direction of the presented research topic at least 1-2 lines. Also the commercial aspects of how patients education and community can benefit.  Most of the sections discussed lack citing the references from the latest and previous studies. Below are a few suggestions the authors are requested to discuss and cite in appropriate sections.

Canadian journal of neurological sciences 35, no. 1 (2008): 8-21

Marine Drugs 18, no. 4 (2020): 201.

Neuromuscular complications of statins." Physical medicine and rehabilitation clinics of North America 19, no. 1 (2008): 47-59.

Marine drugs 18, no. 4 (2020): 226.

Current Neurology and Neuroscience Reports 20, no. 10 (2020): 1-7.

International journal of nanomedicine 10 (2015): 321.

Author Response

Q The topic submitted has already large reviews published earlier but does cover the latest significant research data to the existing field of research of statins. The article is not very well articulated and needs English language revisions and even formatting of the manuscript as per the MDPI guidelines.The manuscript needs to be checked for proper numbering and citation of references.

A We have followed the reviewer’s suggestions better ordering the list of references and citations and reviewing english language

Q The introduction needs to be enlarged with more background.

A We have included some more concepts in the introduction; in addition, we have specified the type of databases search to select the papers included in the review.

Q Abstract and conclusion should include a sentence proposing the future direction of the presented research topic at least 1-2 lines. Also the commercial aspects of how patients education and community can benefit. 

A Abstract and conclusions have been modified adding some final considerations better addressing the use of statins and improving awareness either of patients or medical doctors about the correct approach

Q Most of the sections discussed lack citing the references from the latest and previous studies. Below are a few suggestions the authors are requested to discuss and cite in appropriate sections.

Canadian journal of neurological sciences 35, no. 1 (2008): 8-21

Marine Drugs 18, no. 4 (2020): 201.

Neuromuscular complications of statins." Physical medicine and rehabilitation clinics of North America 19, no. 1 (2008): 47-59.

Marine drugs 18, no. 4 (2020): 226.

Current Neurology and Neuroscience Reports 20, no. 10 (2020): 1-7. NO

International journal of nanomedicine 10 (2015): 321.

A We have incorporated in the text some relevant concepts coming mainly from Ahn et al (Physical medicine and rehabilitation clinics of North America, 2008) and from Crisan et al (Current Neurology and Neuroscience Reports, 2020)